# LIGHTWEIGHT CROSS-TEXT-VISION PROMPTING DIF-FUSION NETWORK FOR MEDICAL IMAGE SEGMENTA-TION

## ABSTRACT

Accurate segmentation of anatomical and pathological structures is fundamental for reliable medical image analysis. Recently, UNet architectures have achieved remarkable performance in medical image segmentation. However, some challenges still exist: (1) Segmentation masks produced by UNet lack fine-grained details, decaying segmentation results; (2) UNet facilitates multi-scale feature fusion, yet the absence of explicit semantic prompts results in imprecise boundary predictions. To address these issues, we construct a lightweight cross-text-vision prompting diffusion network (LCPDN) to improve medical image segmentation accuracy and robustness. Specifically, we develop a cross-text-vision prompting feature learning (CPRFL) module that enables diffusion models to capture fine-grained representations guided by aligned visual and textual information. To further enhance the performance, a lightweight text-vision fusion representation (LTFR) module is modeled to effectively integrate visual features with diagnostic knowledge. Extensive experiments on multiple public datasets demonstrate that our approach achieves state-of-the-art (SOTA) performance with better generalization, particularly under low-data or noisy conditions, highlighting its potential for medical image segmentation tasks. The code is publicly available at https://anonymous.4open.science/r/segmentation.

## 1 INTRODUCTION

Accurate pixel-level segmentation of specific lesion regions or pathological features is critical in medical image analysis Moghbel et al. (2018). It offers multidimensional decision support for clinicians, such as precise interpretation of lesion heterogeneity, timely optimization of treatment strategies, and prompt interventions. However, due to the inherent complexity and variability of medical images, achieving high-precision automated segmentation remains a great challenge. In this context, UNet Ronneberger et al. (2015), with its encoder-decoder structure and skip connections, effectively extracts deep features and restores high resolution by mitigating the loss of spatial information caused by downsampling. It also significantly improves detail retention and segmentation accuracy, positioning UNet as a cornerstone in the field. With ongoing advancements, several UNet-based models have emerged, including TransUNet Chen et al. (2021), SwinUNet Cao et al. (2022), and Mamba-UNet Wang et al. (2024b), which have substantially improved segmentation accuracy and model robustness, driving progress in medical image segmentation and showing great promise for clinical applications.

Although UNet and its variants have achieved considerable success in numerous applications, noise in medical images still remains, often deteriorating segmentation performance Kazerouni et al. (2022b). To overcome this problem, diffusion models have emerged as a promising solution. These generative models are trained through variational inference and Markov chains, effectively reducing noise and uncertainty in medical images, thus improving segmentation robustness and accuracy Ho et al. (2020). By learning the reverse diffusion process, diffusion models are capable of denoising and enhancing the smoothness of Gaussian-blurred images. These capabilities make the integration of diffusion models into UNet popular, further improving the performance of segmentation.

However, most UNet-based variants that absorb diffusion models are often developed for the segmentation of single-modal images. Although these models achieve advanced performance, they still face a persistent problem: Obtaining expert-level segmentation annotations for medical images is expensive and difficult, leading to a scarcity of high-quality annotations Feng (2024). To address this issue, Huang et al. Huang et al. (2021) utilized radiology reports to study global and local representations by comparing image subregions with textual annotations, while Li et al. Li et al. (2023) introduced medical text annotations in visual transformers. Han et al. Han et al. (2023) adopted text-guided prompt to pretrain encoders on image-text pairs from natural images, extracting more valuable information, and Feng et al. Feng (2024) made it more available to extract knowledge from textual diagnostic information for medical visual representations. Their success highlights the feasibility of incorporating textual diagnostic information into medical image segmentation.

Although text-guided approaches have shown promise in medical image segmentation, key challenges still remain. First, segmentation masks Zhu et al. (2024) generated by encoder–decoder architectures often lack fine-grained details, thereby degrading performance. Second, the effective use of semantic information from low-cost textual diagnostic data to enhance segmentation accuracy is still under-explored. To address these limitations, we aim to align models that incorporate textual diagnostic information into visual–semantic representations, thereby improving overall segmentation outcomes.

Consequently, we propose a lightweight cross-text-vision prompting diffusion network (LCPDN) for medical image segmentation. In many tasks, fine-grained lesion or organ details are often overlooked Xu et al. (2019). LCPDN mitigates this by integrating text and image embeddings to generate image–tag pairs at an early stage, which serve as additional prompts for subsequent feature learning. Within the encoder–decoder framework, aligned text and image information are used as conditional prompts for each UNet block, enabling the capture of detailed features across receptive fields. This design allows LCPDN to preserve both temporal and semantic consistency throughout the pipeline. To ensure prompt accuracy, we introduce a text-based attention mechanism Huemann et al. (2024), where textual embeddings act as queries and image features as keys and values. Furthermore, we develop a lightweight text–vision fusion representation (LTFR) module, built on an MLP backbone, to integrate textual and visual features with fewer dynamic weights, thereby enhancing feature representation. The main contributions are threefold:

- We develop a novel cross-text-vision prompting robust feature learning module to avoid segmentation mask problem. It directs the robust visual feature extraction toward key regions by the aligned vision and semantic prompt, leveraging the intrinsic information propagation of diffusion models to generate accurate representations with fine-grained details.

- We propose a lightweight text-vision fusion representation module that effectively integrates features from different sources, enabling a more comprehensive visual-semantic representation, which in turn enhances segmentation accuracy and performance. Moreover, the lightweight network architecture allows for rapid and efficient feature fusion, demonstrating its strong generalizability.

- We conduct extensive experiments on three datasets, achieving outperforming segmentation results by making comparisons with some state-of-the-art methods, which validate the advantages of the proposed model.

## 2 RELATED WORK

**Medical image segmentation** Early approaches to medical image segmentation were dominated by convolutional neural networks (CNNs), including fully convolutional networks (FCNs) Long et al. (2015) and UNet Ronneberger et al. (2015). UNet, with its encoder–decoder structure and skip connections, effectively mitigates spatial information loss from downsampling and has since been extensively validated and extended. For example, VNet Milletari et al. (2016) and 3D UNet Çiçek et al. (2016) extend segmentation to volumetric data, while nnUNet Isensee et al. (2018) provides an automated and robust baseline across diverse datasets. To capture complex contextual relationships and fine-grained structures, attention UNet Oktay et al. (2018) introduces attention mechanisms, and more recent Transformer-based models Cao et al. (2022); Li et al. (2023); Yang et al. (2022); Hu et al. (2023b); Wu et al. (2024a) leverage self-attention to model long-range dependencies and global context. Building on these advances, studies incorporating adversarial training Xue et al. (2018), domain adaptation Kamnitsas et al. (2017), and self-supervised learning Zhou et al. (2019) have

further addressed challenges of limited data, cross-domain generalization, and segmentation under complex pathological conditions. Collectively, these developments not only improve segmentation accuracy but also introduce strategies to mitigate noise and modality heterogeneity.

**Diffusion Models in Medical Image Segmentation** Denoising diffusion probabilistic models Ho et al. (2020) and score-based generative models using stochastic differential equations Song et al. (2020) has established the theoretical underpinnings for this framework. In medical image segmentation, diffusion models have been primarily explored in two directions. First, they model and remove noise through the diffusion process, thereby enhancing the fidelity of segmentation masks. Second, conditional diffusion incorporates prior—anatomical context—into the generative process to improve segmentation performance. Dhariwal et al. Dhariwal & Nichol (2021a) proposed a conditional diffusion network that integrates auxiliary contextual cues (e.g., lesion location prompts) to guide the reverse diffusion, thereby preserving global structure while enhancing local detail Wu et al. (2024b).Ma Ma (2025) introduced cross-modal embeddings as conditional inputs for denoising, while incorporating timestep information as an independent input to capture the transitional noise distribution of label embeddings, thereby guiding the denoising process and enhancing 3D segmentation performance.Furthermore, hybrid approaches that integrate diffusion models into UNet architectures Wang et al. (2025); Hu et al. (2023a) have demonstrated strong robustness on imaging scenarios, such as low-contrast or noise-contaminated medical scans. Moreover, the further incorporation of uncertainty and noise has empowered diffusion models to dynamically adjust the denoising in response to the characteristics of input, thereby achieving more refined and detail-preserving segmentation results Tang et al. (2023a); Kazerouni et al. (2022a).

**Multimodal fusion methods** In clinical practice, textual annotations provide semantic cues valuable for segmentation Huang et al. (2021). Initially, multimodal fusion methods rely on simple feature concatenation or weighting to achieve joint representation of image and text data Xu (2019). Later, vision-language models such as CLIP Li et al. (2023) have introduced novel strategy for cross-modal alignment. Thus, several studies have integrated pretrained text encoders (e.g., BioBERT Lee et al. (2020)) with image encoders to create Transformer-based fusion networks. Feng proposed a label-supervised medical image segmentation method, which reduces reliance on pixel-level annotations by incorporating additional expert and medical textual knowledge Feng (2024). Huang et al. leveraged cross-modal masked features to focus on key feature tokens between image and text, while mitigating the interference of background from both modalities Huang et al. (2024). Furthermore, recent work Liu et al. (2023); Xie et al. (2024); Shah et al. (2024); Chng et al. (2024) has introduced sophisticated feature fusion modules. Liu et al. proposed both a multimodal mutual attention module and an iterative multimodal interaction module, facilitating continuous and in-depth interactions between language and visual features. Shah et al. developed LQMFormer, which incorporates a Gaussian-enhanced multimodal fusion module to enhance overall representation by extracting rich local and global visual-language relationships Shah et al. (2024). These advances have marked significant progress in multimodal fusion.

## 3 METHOD

In this section, we first outline the overall framework of the proposed LCPDN (Section 3.1), followed by brief descriptions of the image and text encoders derived from pretrained vision–language models, including CLIP Radford et al. (2021) and BioBERT Alsentzer et al. (2019). We then detail two core components of the framework: the cross-text-vision prompting robust feature learning (CPRFL) module (Section 3.3) and the lightweight text–vision fusion representation (LTFR) module (Section 3.4). CPRFL injects conditional prompts, generated by a text–vision alignment block, into each UNet layer to capture fine-grained features across multiple receptive fields, whereas LTFR fuses heterogeneous features to strengthen the semantic representation of visual information.

### 3.1 OVERALL FRAMEWORK

Given a pair of image and text as input, a deep neural network is modeled to well segment the specific anatomical structures or pathological regions of clinical interest. Our training dataset can be formalized as $\mathcal{D} = \{(I^1, T^1), (I^2, T^2), \ldots, (I^N, T^N)\}$,where $I$ denotes medical images and $T$ their corresponding textual annotations. We further study how to extract task-relevant visual features

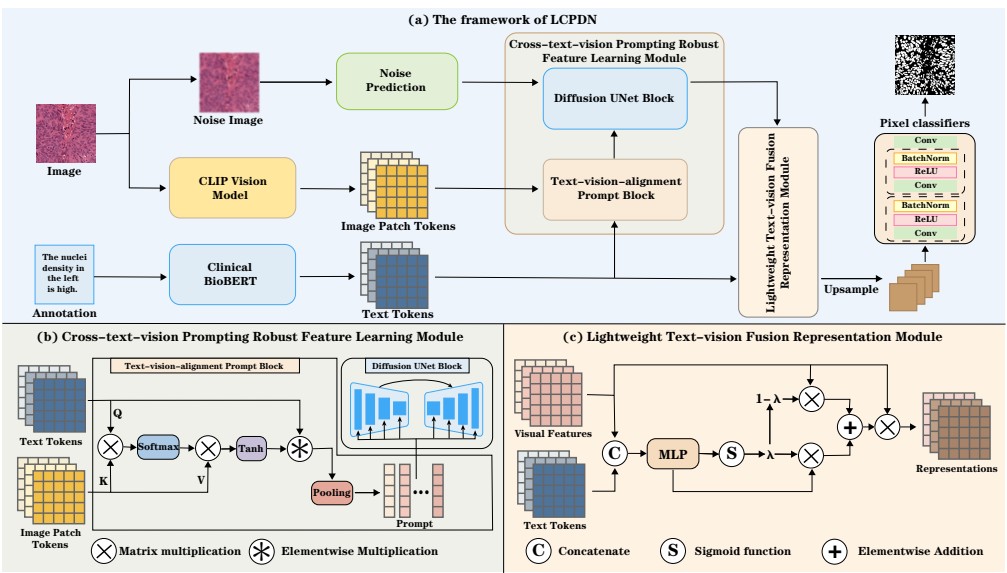

Figure 1: (a) Overview of our segmentation framework, which combines noisy images with conditional prompts to ensure that information from text-vision aligned prompts is perceived at each step. (b) The proposed CPRFL module, which integrates the visual foundation and semantic richness to produce the input prompts. (c) The proposed LTFR module, where we align the two feature types and apply a dynamic fusion strategy to enhance semantic representation.

using diffusion models by focusing on target regions. To achieve this, we integrate text vectors with image embeddings in advance to form guidance prompts, which are then fed into the visual training network Baranchuk et al. (2021). This mechanism helps the model locate critical regions more explicitly, thereby enabling the extraction of more effective visual representations. As illustrated in Fig. 1, we propose a multi-branch architecture, which extracts visual and textual representations via a combination of a UNet–based diffusion model Dhariwal & Nichol (2021b) and a pretrained clinical language encoder (BioBERT). This design enables alignment between the textual diagnostic information and the Markovian steps of the reverse diffusion process. Each modality is first processed by a pretrained backbone to extract multimodal feature representation. These features are then fused using our proposed LTFR module, followed by a pixel-wise classifier to generate the final segmentation predictions.

## 3.2 FEATURE EXTRACTION

To enable text-guided medical image segmentation and to reinforce the semantic consistency between visual and textual representations, we employ pretrained encoders derived from CLIP and Clinical BioBERT to extract informative visual and textual features, respectively. Given a medical image $I$ and its associated diagnostic report $T$, the encoders produce corresponding visual features $F^I$ and textual features $F^T$, as detailed below.

**Image encoder** For a given input image $I$, we adopt the vision encoder from CLIP, which is a Transformer-based architecture. The encoder is used in a frozen state to extract visual features. The input image is tokenized with a prepended token that captures global semantic information. Following standard practice in prior work Wang et al. (2022b); Lüddecke & Ecker (2022), we extract the global representation from the token. The resulting visual representation is denoted as $F^I \in \mathbb{R}^{B \times M \times D}$, where $B$ is the batch size, $M$ denotes the number of image patches, and $D$ is the dimensionality of each patch embedding.

**Text encoder** As described earlier, diagnostic reports are naturally paired with medical images during acquisition and can be incorporated into the model without additional annotation costs. Owing to their compactness and semantic richness, these textual descriptions serve as a valuable complement

to visual data. As illustrated in Fig. 1, we utilize Clinical BioBERT, trained on the MIMIC-III dataset, to encode the diagnostic text and extract clinically-relevant representations Johnson et al. (2016). Given a textual input $T$, we obtain feature vectors through the BioBERT backbone. They are subsequently injected into the reverse diffusion process via a cross-modal attention mechanism aligned with intermediate activations across Markov steps. We denote the textual representation as $F^T \in \mathbb{R}^{B \times N \times D}$, where $N$ represents the number of tokens in the input text and $D$ is the dimensionality of each token embedding.

### 3.3 CROSS-TEXT-VISION PROMPTING ROBUST FEATURE LEARNING MODULE

CPRFL module is developed to learn robust visual representations from medical images with conditional prompts. It consists of both a text-vision-alignment prompt block and a diffusion UNet block, by which the conditional prompts and robust features can be learned, respectively.

**Text-vision-alignment prompt block**   Text-vision-alignment prompt block integrates both textual and visual tokens to generate the final conditional prompt. As illustrated in Fig. 1, we introduce a language–vision prompt fusion mechanism that injects semantic representations from the text into visual features through a cross-attention strategy. Inspired by the principle of pixel-level attention, this approach is adapted to operation on the global representations extracted by CLIP. The resulting fused representation is aligned with the textual semantics while remaining grounded in the visual context of the input image. After removing the batch dimension, we compute the attention between the projected textual vectors $\mathbf{Q} \in \mathbb{R}^{N \times D}$ and the projected visual features $\mathbf{K}, \mathbf{V} \in \mathbb{R}^{M \times D}$ where $N$ denotes the number of textual tokens and $M$ represents the number of visual patches. The resulting attention map is defined as follows:

$$\mathbf{A} = \text{softmax}\left( \frac{(\mathbf{Q}\mathbf{W}^Q)(\mathbf{K}\mathbf{W}^K)^\top}{\sqrt{D}} \right),$$
$$\mathbf{Z} = \mathbf{A} \cdot (\mathbf{V}\mathbf{W}^V), \tag{1}$$

where $\mathbf{W}^Q, \mathbf{W}^K, \mathbf{W}^V \in \mathbb{R}^{D \times D}$ is a learnable projection matrix that maps the original feature space to a shared latent dimension $D$.

To reinforce the alignment between semantic text vectors and visual contextual features, we introduce a nonlinear fusion mechanism based on element-wise interactions. Specifically, the attended visual response $\mathbf{Z}$ is combined with the projected textual queries $\mathbf{Q}\mathbf{W}^Q$ through element-wise multiplication (denoted by $\odot$), followed by a hyperbolic tangent (tanh) activation to normalize the fused representation and emphasize salient cross-modal interactions. The resulting output $\mathbf{Q}^* \in \mathbb{R}^{N \times D}$ captures the integrated representation of these two modalities.

$$\mathbf{Q}^* = \tanh(\mathbf{Z}) \odot \left(\mathbf{Q}\mathbf{W}^Q\right). \tag{2}$$

Finally, the fused feature vectors are aggregated along the token dimension to derive a unified representation,

$$\mathbf{E} = \frac{1}{N} \sum_{n=1}^{N} \mathbf{Q}_n^*. \tag{3}$$

The resulting embedding $\mathbf{E} \in \mathbb{R}^D$ integrates both the visual grounding and semantic richness of the input prompts. By combining image and text tokens, the prompt ensures that the diffusion model generates features that are both temporally consistent and semantically aligned. After obtaining $\mathbf{E}$, we employ a fusion strategy to incorporate the prompt into the temporal embeddings of the diffusion model, thereby facilitating prompt-driven robust visual feature extraction. Specifically, We employ a linear transformation parameterized by a projection matrix $\mathbf{W}_t \in \mathbb{R}^{D \times d}$ and a bias term $\mathbf{b}_t \in \mathbb{R}^d$, mapping the text vector $\mathbf{E} \in \mathbb{R}^D$ into the same latent dimension $d$ as the temporal embedding.

$$\text{TextProj}(\mathbf{E}) = \mathbf{E} \cdot \mathbf{W}_t + \mathbf{b}_t. \tag{4}$$

Subsequently, the mapped result is added to the temporal embedding $\mathbf{M} \in \mathbb{R}^d$ corresponding to the time step $t$, forming the fused prompt vector $\mathbf{P}$, which serves as the conditional input to the downstream neural network module.

$$\mathbf{P} = \mathbf{M} + \text{TextProj}(\mathbf{E}) = \mathbf{M} + (\mathbf{E} \cdot \mathbf{W}_t + \mathbf{b}_t). \tag{5}$$

This approach ensures that the model is able to perceive information from the conditional prompt at each time step, effectively enhancing the semantic consistency of the generated features.

**Diffusion UNet block** The diffusion UNet comprises downsampling and upsampling blocks with skip connections, enhanced by cross-attention modules Vaswani et al. (2017) to align conditional prompts with spatial features across UNet, and self-attention mechanisms along the temporal axis to enrich contextual understanding. Each UNet block therefore produces feature maps to encode both semantic alignment and temporal coherence. To obtain latent representation within UNet for medical image understanding, it takes two inputs: a noisy image $\mathbf{N}_{img}$ and a conditional prompt $\mathbf{P}$. In our framework, the noisy image helps preserve the fidelity of generated features during the diffusion process, while the conditional prompt integrates both the reference text and an image embedding. The reference text ensures temporal consistency across visual representations, whereas the image embedding serves as a complementary condition to enhance the richness and granularity of extracted features.

$$\mathbf{F}_{img} = f_{DUN}\left(\mathbf{N}_{img}, \mathbf{P}\right), \tag{6}$$

where $f_{DUN}$ is the nonlinear mapping of UNet. After visual feature extraction via the diffusion UNet, we obtain the fused images $\mathbf{F}_{img}$ for subsequent network processing.

### 3.4 LIGHTWEIGHT TEXT-VISION FUSION REPRESENTATION MODULE

Here, we address the alignment issue of textual and visual features at different scales. To effectively leverage the multimodal feature information extracted and fused in the previous steps, we propose a multi-scale cross-modal feature fusion module, termed lightweight text-vision fusion representation (LTFR). Specifically, through the aforementioned process, we obtain two multimodal feature streams: textual features $\mathbf{F}_{text}$ by BioBERT and the fused images $\mathbf{F}_{img}$ extracted by CPRFL. Here, $\mathbf{F}_{text}$ corresponds to the textual representation $F^T$ obtained from the text encoder described earlier.

For these multimodal data, visual features extracted from hierarchical layers with different spatial resolutions and textual descriptions often present inconsistencies in dimensionality and structural form. To circumvent these problems, we first perform dimensional alignment and positional mapping for both modalities, followed by information interaction using a dynamic fusion strategy. Let $\mathbf{F}_{img} \in \mathbb{R}^{1 \times M \times D}$ represent the image features and $\mathbf{F}_{text} \in \mathbb{R}^{1 \times N \times D}$ represent the textual features. On the condition that $N \neq M$, $\mathbf{F}_{text}$ is either replicated or truncated along the spatial dimension to obtain an aligned textual feature $\mathbf{F}_{text}^{\uparrow} \in \mathbb{R}^{1 \times M \times D}$. The features $\mathbf{F}_{img}$ and $\mathbf{F}_{text}^{\uparrow}$ are then concatenated along the feature dimension to facilitate cross-modal interaction.

$$\mathbf{F}_{cat} = \text{Concat}\left(\mathbf{F}_{img}, \mathbf{F}_{text}^{\uparrow}\right), \quad \mathbf{F}_{cat} \in \mathbb{R}^{1 \times M \times 2D}, \tag{7}$$

where the fused feature $\mathbf{F}_{cat}$ is input into a multilayer perceptron (MLP) Wang et al. (2024a) with nonlinear activation functions to extract cross-modal interaction features with fewer parameters:

$$\mathbf{F}_{fuse} = \text{MLP}\left(\mathbf{F}_{cat}\right), \quad \mathbf{F}_{fuse} \in \mathbb{R}^{1 \times M \times D}. \tag{8}$$

To further optimize the weights between modalities, we introduce a dynamic mechanism to compute the proportion of visual and semantic information within the fused features. Specifically, the Sigmoid activation function is suggested to generate position-dependent weights $\lambda \in [0, 1]$, which are integrated according to the dynamic weighting mechanism:

$$\mathbf{F}_{fuse}^{*} = \lambda \cdot \mathbf{F}_{fuse} + (1 - \lambda) \cdot \mathbf{F}_{img}. \tag{9}$$

The reweighted features are then augmented with visual information to obtain the final fused cross-modal feature representation.

$$\mathbf{H}_{fuse} = \mathbf{F}_{fuse}^{*} \cdot \mathbf{F}_{img}. \tag{10}$$

Considering the multi-scale features extracted by the visual backbone, we apply the proposed fusion module at each scale to jointly encode local and global textual semantics. The fused features $\mathbf{H}_{fuse}$ from all scales are then upsampled to a uniform spatial resolution and concatenated along the channel dimension to form the final multimodal representation:

$$\mathbf{H} = \text{Concat}\left(\text{Upsample}(\mathbf{H}_{fuse1}), \dots, \text{Upsample}(\mathbf{H}_{fuseS})\right), \tag{11}$$

where $S$ represents the number of fusion scales. The resulting representation $\mathbf{H}$ effectively integrates multi-level image structures with textual semantic information, providing enhanced semantic support for downstream multimodal understanding tasks.

## 4 EXPERIMENTS

### 4.1 DATASETS AND METRICS

**Datasets**   We evaluate the performance of our method using three publicly available datasets:

1) MoNuSeg Kumar et al. (2019): Released as part of the MICCAI 2018 MoNuSeg Challenge, this dataset consists of Hematoxylin and Eosin (H&E)-stained tissue images with more than 21,000 meticulously annotated nuclear boundaries. In the experiments, we used a reduced subset, selecting only 5 images from the original training set while retaining the 14 original test images.

2) QaTa-COVID19 Degerli et al. (2022): Created by Tampere University and Qatar University, this dataset comprises 121,378 chest X-ray (CXR) images, including 9,258 confirmed COVID-19 cases, each paired with a corresponding ground truth mask. In the experiments, we select 150 images for training and 50 for testing as Li et al. (2023).

3) MosMedData+ Morozov et al. (2020): This dataset comprises 2,729 computed tomography (CT) slices depicting lung infections. In the experiments, we select 150 images for training and 30 for testing. Following Huang et al. (2024), we incorporate their extended textual annotations to enhance the vision–language model.

**Metrics**   We employ Dice similarity coefficient and Intersection over Union (IoU) metric for performance evaluation, which are defined as follows:

$$Dice = \frac{2 * TP}{(TP + FN) + (TP + FP)}, \tag{12}$$

$$IoU = \frac{TP}{TP + FN + FP}, \tag{13}$$

where $TP$, $FN$ and $FP$ denote true positives, false negatives and false positives, respectively.

### 4.2 IMPLEMENT DETAILS

In our experiments, we adopt the Adam Kingma & Ba (2014) optimizer with a fixed learning rate of 1e-4. All experiments are run using PyTorch 1.10.0 with CUDA 11.3, and the training is performed on a single NVIDIA GeForce RTX 3080Ti GPU with 12 GB memory. We set the batch size to 1 and train each model for 50 epochs. All input images are resized to 256×256 pixels. For the UNet decoder, the intermediate blocks $B$ are configured as {6, 8, 12, 16} for MoNuSeg, {4, 6, 8, 12} for QaTa-COVID19, and {4, 6, 8, 12} for MosMedData+, corresponding to diffusion steps $t = \{50, 150, 250\}$, respectively. The feature dimensions across each layer of the UNet encoder-decoder architecture are set to {1024, 512, 512, 256}. Bilinear interpolation is applied for upsampling, and the shared noise is introduced in the diffusion process to enhance training stability.

### 4.3 BASELINE

In the experiments, we adopt two categories of methods for comparison: **A)** Classical single-modality medical segmentation methods: (1) UCTransNet Wang et al. (2022a), introducing a Channel Transformer (CTrans) to replace conventional skip connections; (2) MGCC Tang et al. (2023b), presenting a multi-level global context cross-consistency (MGCC) framework that leverages images synthesized by latent diffusion models to augment unlabeled data for semi-supervised learning. (2) MedSegDiff Wu et al. (2024c), which combines diffusion probabilistic modeling with transformer-based architectures to improve feature interaction and achieve more accurate medical image segmentation across diverse modalities. **B)** Text-driven medical segmentation methods: (1) LViT Li et al. (2023), a multimodal segmentation framework that enhances Transformer-based visual representations by incorporating medical text annotations; (2) RecLMIS Huang et al. (2024), which employs conditional contrastive learning and cross-modal reconstruction to achieve fine-grained alignment between modalities; (3) TextDiff Feng (2024), a label-efficient segmentation method that leverages expert knowledge from medical text annotations to reduce the reliance of diffusion models on pixel-level labels. (3) C2FVL Shan et al. (2023), which proposes a coarse-to-fine vision–language alignment framework that integrates lesion-specific textual cues with image features to enhance COVID-19 lesion segmentation on chest X-ray and CT data.

Table 1: Quantitative segmentation results on MoNuSeg,QaTa-COVID19 and MosMedData+ dataset

| Method | Param(M) | MoNuSeg | | QaTa-COVID19 | | MosMedData+ | |
|---|---|---|---|---|---|---|---|
| | | Dice(%)↑ | IoU(%)↑ | Dice(%)↑ | IoU(%)↑ | Dice(%)↑ | IoU(%)↑ |
| LViT | 29.71 | 75.41 | 60.83 | 71.69 | 60.41 | 55.19 | 42.43 |
| UCTransNet | 66.24 | 78.25 | 64.44 | 62.41 | 50.18 | 55.30 | 41.72 |
| MGCC | 122.18 | 73.72 | 58.86 | 64.42 | 51.64 | 48.44 | 36.74 |
| RecLMIS | 23.7 | 76.28 | 61.94 | 71.15 | 59.72 | 58.94 | 44.56 |
| TextDiff | 9.68 | 78.67 | 64.98 | 71.28 | 59.27 | 66.58 | 52.12 |
| MedSegDiff | 46.0 | 48.98 | 35.34 | 44.71 | 32.33 | 29.68 | 18.31 |
| C2FVL | 24.0 | 72.14 | 56.91 | 71.60 | 59.96 | 44.00 | 31.14 |
| **LCPDN (Ours)** | **4.52** | **79.74** | **66.45** | **73.47** | **61.43** | **68.97** | **54.00** |

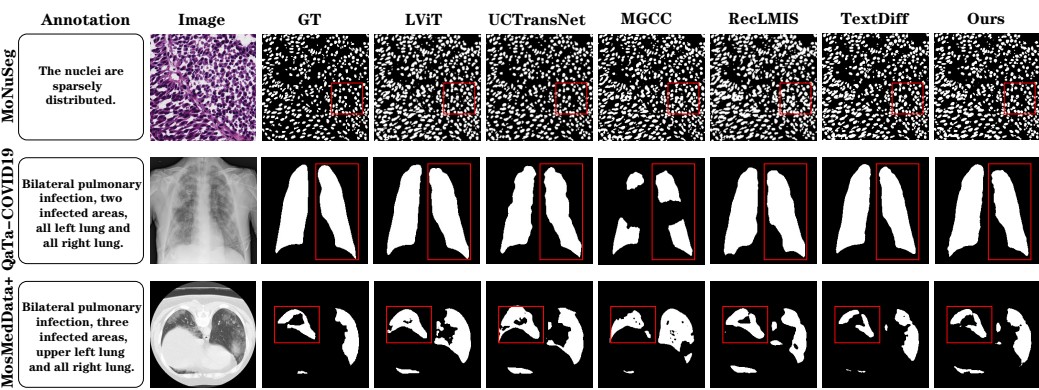

Figure 2: Visual segmentation comparisons with some methods on the MoNuSeg,QaTa-COVID19 and MosMedData+ dataset. As observed, Ours achieves superior performance with detailed anatomical information.

## 4.4 RESULTS

**Performance evaluation** We conduct experiments on MoNuSeg, QaTa-COVID19, and MosMed-Data+ datasets. For a fair comparison, we retrain all comparison methods using their default hyperparameters under identical experimental conditions and report their best performance. Table 1 presents a detailed performance comparison between our method and the competing approaches, from which it can observe the superiority of the proposed method either from Dice, IoU, or from parameter volume. Namely, the proposed model is a lightweight network with better performance.

**Visualization of segmentation** Figure 2 shows the segmentation results of our method compared with baseline approaches across three datasets. Relative to the ground truth, text-driven vision methods consistently surpass single-modality segmentation approaches, particularly when trained on larger datasets. This highlights the value of integrating textual information into medical image segmentation, aligning with our motivation to leverage multimodal cues for performance gains. Our method further achieves segmentations with fewer errors and closer resemblance to the ground truth, benefiting from its ability to integrate multi-scale image features with semantic information embedded in medical text. The consistent improvements observed across three datasets demonstrate both the robustness and generalizability of the proposed framework.

## 4.5 ABLATION STUDY

We conduct ablation study on MoNuSeg, QaTa-COVID19, and MosMedData+ datasets, using images of size 256 × 256. At first, we conduct ablation experiments by removing CPRFL from our model, and the corresponding results are shown in Table 2. The removal of CPRFL leads to a significant

Table 2: Ablation study on MoNuSeg, QaTa-COVID19 and MosMedData+. ✓/✗ indicates that the module is used or not.

| CPRFL | LTFR | MoNuSeg | | QaTa-COVID19 | | MosMedData+ | |
|---|---|---|---|---|---|---|---|
| | | Dice(%) | IoU(%) | Dice(%) | IoU(%) | Dice(%) | IoU(%) |
| ✗ | ✗ | 79.35 | 65.96 | 68.42 | 54.94 | 63.5 | 48.67 |
| ✓ | ✗ | 79.29 | 65.90 | 71.00 | 58.45 | 65.91 | 51.19 |
| ✗ | ✓ | 79.70 | 66.38 | 72.42 | 60.71 | 65.86 | 51.36 |
| ✓ | ✓ | 79.74 | 66.45 | 73.47 | 61.43 | 68.97 | 54.00 |

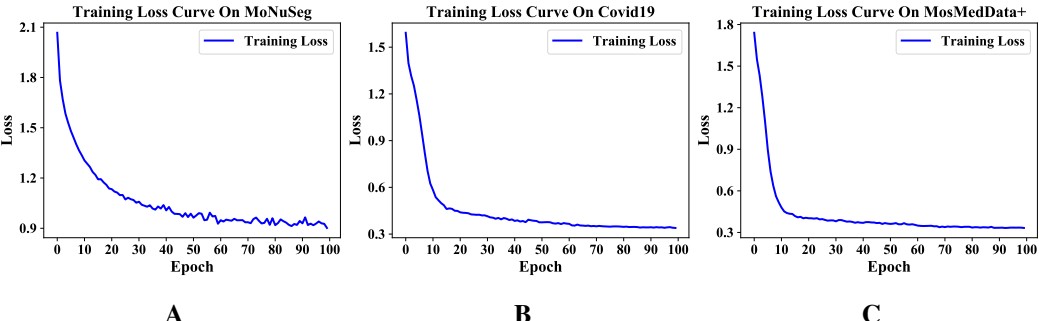

Figure 3: Convergence analysis. **A.** The loss curve for MoNuSeg dataset. **B.** The loss curve for QaTa-COVID19 dataset. **C.** The loss curve for MosMedData+ dataset.

degradation in segmentation performance. Namely, CPRFL plays a crucial role in enhancing the semantic consistency of the generated features, thereby improving segmentation accuracy. In addition, since multimodal fusion remains essential, we replace LTFR with a standard cross-attention module for comparison. As shown in Table 2, this substitution results in decreases of up to 3.06% and 2.98% in Dice and IoU scores, respectively, indicating that LTFR effectively integrates multimodal features and enhances contextual feature representations.Overall, although the improvements on the smaller MoNuSeg dataset are relatively few, our method on the larger QaTa-COVID19 and MosMedData+ datasets achieves significant increases of 5.05% and 5.47% in Dice scores, 6.49% and 5.33% in IoU scores, respectively, demonstrating its innovation and practical effectiveness.

### 4.6 CONVERGENCE

To further examine the convergence of our framework, we plot the total segmentation loss over training iterations on three datasets. As illustrated in Figure 3, the proposed model demonstrates a steady convergence trend with minimal fluctuations as training progresses.

## 5 CONCLUSION

In this work, we introduce a novel medical image segmentation method that integrates pretrained vision-language models with diffusion models. The method redefines the segmentation task as a text-prompt-driven fine-grained problem. By incorporating a novel cross-text-vision prompting robust feature learning module and a lightweight text-vision fusion representation module, we effectively capture detailed information from key regions in medical images. Our prompt-motivated segmentation framework embeds a diffusion model into UNet architecture and leverages vision and text encoders from pretrained models such as CLIP and BioBERT to achieve precise alignment and fusion of cross-modal information. This enhances segmentation performance in terms of noise removal and detail recovery. Experimental results show that, compared to state-of-the-art methods, our model significantly improves performance across multiple medical imaging datasets.

## REPRODUCIBILITY STATEMENT

The code is publicly available at `https://anonymous.4open.science/r/segmentation`. The dataset preparation and preprocessing follow LViT Li et al. (2023), whose publicly available GitHub repository provides detailed implementation of the data processing steps.To ensure the reproducibility and completeness of this work, we provide an appendix consisting of two main parts. Appendix A presents visualizations of additional baseline methods, while Appendix C offers further qualitative segmentation results on the three datasets used in our experiments, allowing a clearer comparison of the segmentation performance across different approaches.

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

## A   VISUALIZATION OF ADDITIONAL METHODS

**Visualization of segmentation**   Figure 4 presents a comparison of segmentation results between our method and two baseline approaches across three datasets. In general, text-driven vision methods consistently outperform single-modality segmentation approaches, aligning with our motivation to leverage textual information for enhanced segmentation. By contrast, the performance of MedSegDiff is less satisfactory, which may be attributed to its reliance on large-scale data and stable sampling through multiple iterations. On datasets with limited samples or image modalities that differ signifi­cantly from the experimental conditions reported in its original study, its performance tends to fall short of more robust conventional architectures.

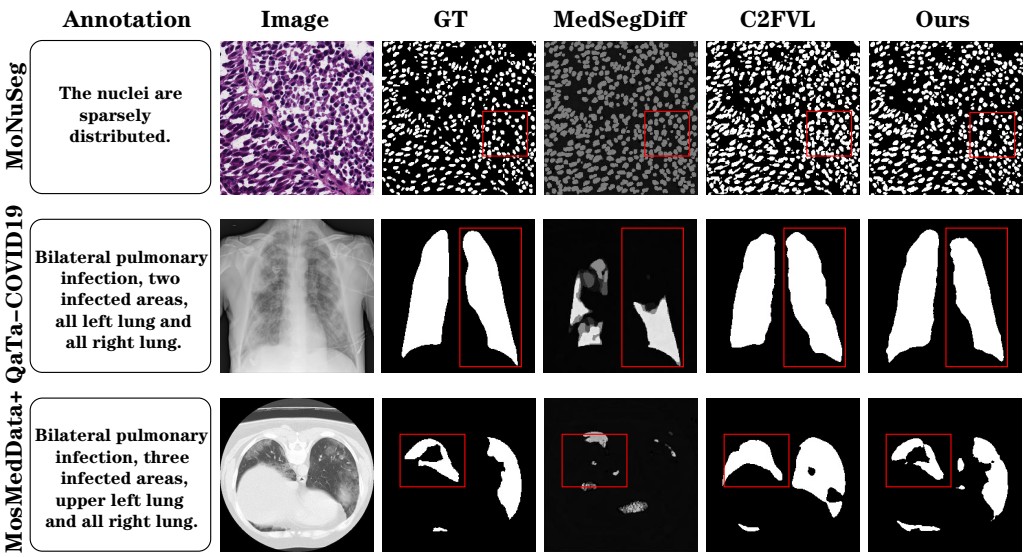

Figure 4: Visual segmentation comparisons with other methods on the MoNuSeg,QaTa-COVID19 and MosMedData+ dataset.

## B   LLMS DISCLAIMER

In this paper, large language models (LLMs) were employed solely for post-completion review and refinement of the manuscript, with the aim of improving its fluency and overall readability. Specifically, LLMs were used to check for consistency in terminology throughout the paper and to suggest more precise domain-specific expressions where appropriate, thereby enhancing the professionalism of the final presentation.

