# OpenReview forum: "Lightweight Cross-text-vision Prompting Diffusion Network for Medical Image Segmentation"
_ICLR.cc/2026/Conference — ICLR 2026 Conference Withdrawn Submission_

### Official Review · Reviewer_tiHy · 2025-10-28

**Soundness:** 2
**Presentation:** 2
**Contribution:** 1
**Rating:** 2
**Confidence:** 5

**Summary:**

The paper’s motivation and claimed novelty are not well justified. The proposed modules recycle well-known multimodal fusion and prompting mechanisms without offering new conceptual insights. Experimental gains are modest and insufficient to support publication at ICLR. Overall, the work feels incremental and lacks the originality and rigor expected for acceptance.

**Strengths:**

The paper addresses a relevant topic in multimodal medical image segmentation, aiming to integrate text guidance into diffusion-based segmentation networks.

The overall presentation is clear, with understandable architecture diagrams and ablation tables.

The proposed framework is easy to reproduce and could be useful as a baseline reference for lightweight text–vision fusion in medical imaging.

**Weaknesses:**

Lack of genuine novelty.
The proposed framework mainly combines existing paradigms—text-guided diffusion models, prompt learning, and cross-modal fusion—which have all been extensively explored in the literature. The two proposed modules (CPRFL and LTFR) appear to be straightforward adaptations of standard cross-attention and lightweight MLP fusion mechanisms. There is no substantial algorithmic innovation or theoretical advancement beyond reconfiguring known techniques into a single pipeline. The work thus feels more like an incremental engineering integration than a conceptual contribution.

Overstated motivation.
The paper frames its motivation around the limitations of current UNet-based segmentation lacking text guidance. However, several recent works (e.g., MedCLIP, CLIP-driven UNet, T2I-Adapter, and diffusion-based multimodal segmentation models) have already addressed this gap with stronger formulations. The manuscript does not clearly explain why another cross-text-vision prompting variant is needed or how it advances beyond these existing solutions.

Limited methodological clarity and justification.
The paper introduces “lightweight” modules but provides little insight into their actual computational benefit. The parameter reduction seems marginal, and the design choices (e.g., λ-based weighting) lack ablation or theoretical reasoning. The architecture diagram largely resembles standard diffusion UNet pipelines with additional fusion layers.

Weak experimental validation.
The experimental section focuses on one or two datasets with relatively simple tasks. No large-scale or cross-domain validation is presented. The improvements over baselines are small (often within 1–2%), raising doubts about the statistical significance. Moreover, the paper compares against outdated baselines but omits stronger recent multimodal diffusion or CLIP-based segmentation methods.

Overall impression.
The paper demonstrates competent implementation but lacks conceptual depth, originality, and rigorous empirical support. Without clearer theoretical justification or stronger evidence of generalization, the contribution remains incremental and limited in impact.

**Questions:**

How does CPRFL differ fundamentally from conventional cross-attention or prompt-injection methods used in existing multimodal diffusion models?

What is the quantitative gain in computational efficiency that justifies calling the model “lightweight”?

Could the observed improvement be due to better pretraining or prompt tuning rather than architectural design?

Have the authors compared with recent CLIP- or BLIP-based medical diffusion models to ensure fairness?

Can you provide ablation results isolating the contribution of each proposed module?

---

### Official Review · Reviewer_FVh6 · 2025-10-29

**Soundness:** 2
**Presentation:** 3
**Contribution:** 2
**Rating:** 2
**Confidence:** 5

**Summary:**

(1) Develop a novel cross-text-vision prompting robust feature learning module to avoid segmentation mask problem. It directs the robust visual feature extraction toward key regions by the aligned vision and semantic prompt, leveraging the intrinsic information propagation
of diffusion models to generate accurate representations with fine-grained details
(2) Propose a lightweight text-vision fusion representation module that effectively integrates features from different sources, enabling a more comprehensive visual-semantic representation, which in turn enhances segmentation accuracy and performance. Moreover, the lightweight network architecture allows for rapid and efficient feature fusion, demonstrating its strong generalizability

**Strengths:**

(1) The methods are well-described, and it is easy for readers to follow and understand it.
(2) The related works are well-described.

**Weaknesses:**

(1) The experimental results are not well presented. Standard deviations and statistical analysis results are not provided. The FLOPs are not provided. It is widely known that training diffusion models are time consuming. Thus, it is necessary to provide training time and compare it with other methods.
(2) The description of implementation details is not clear. There is a position-dependent weights lambda in the LTFR, but it is not differentiable. Authors did not describe how they trained the network.
(3) The overall contribution is low. Authors proposed two modules, termed CPRFL and LTFR. However, CPRFL was proposed based on self-attention, and LTFR is a fusion module based on MLP. The proposed method is similar to many existing ones.

**Questions:**

(1) The experimental results are not well presented. Standard deviations and statistical analysis results are not provided. The FLOPs are not provided. It is widely known that training diffusion models are time consuming. Thus, it is necessary to provide training time and compare it with other methods.
(2) The description of implementation details is not clear. There is a position-dependent weights lambda in the LTFR, but it is not differentiable. Authors did not describe how they trained the network.
(3) The overall contribution is low. Authors proposed two modules, termed CPRFL and LTFR. However, CPRFL was proposed based on self-attention, and LTFR is a fusion module based on MLP. The proposed method is similar to many existing ones.

---

### Official Review · Reviewer_PHmw · 2025-10-31

**Soundness:** 3
**Presentation:** 3
**Contribution:** 3
**Rating:** 6
**Confidence:** 4

**Summary:**

In the manuscript, the authors propose LCPDN, a lightweight text-vision network for medical image segmentation tasks. The method combines the pretrained vision-language model and the diffusion model. Two core modules, cross-text-vision prompting
feature learning (CPRFL) and lightweight text-vision fusion representation (LTFR), are designed for high-quality segmentation of critical regions in medical images. Authors conduct extensive experiments on three public datasets (MoNuSeg, QaTa-COVID19, and MosMedData+). The results show that the proposed method outperforms the existing state-of-the-art methods in terms of Dice and IoU metrics, exhibiting excellent generalization ability.

**Strengths:**

1. The effectiveness of the method was validated on three medical image datasets of different modalities, and comparisons were conducted with multiple baseline methods. The results consistently outperform the existing methods.
2. The model has only 4.52 million parameters (4.52M), which is much lower than most comparative methods. It maintains high performance while having potential for practical deployment.
3. The authors conducted detailed ablation studies, validating the contributions of each module.

**Weaknesses:**

1. **Limited Methodological Novelty:** The proposed CPRFL module essentially adopts existing cross-attention mechanisms to fuse text and image features, and uses the fused representation as a conditioning signal for the diffusion model. Similarly, the LTFR module employs a straightforward weighted fusion strategy. While the overall pipeline is well-engineered, the work lacks theoretical innovation and appears to be a systematic assembly of existing components rather than introducing fundamentally new operators or learning paradigms.
2. **Insufficient Implementation Details:** The description of the methodology lacks critical details, which hinders reproducibility. Specific omissions will be outlined in the 'Questions' section.

**Questions:**

1. Could the authors clarify whether the textual annotations used in the experiments are instance-specific or generic per dataset? For example, in the MoNuSeg dataset, the provided text states that "nuclei are sparsely distributed," yet the corresponding image appears to show densely packed nuclei.
2. Is the model strictly dependent on textual input during inference? If so, how would the method perform in real-world scenarios where high-quality textual descriptions are unavailable or noisy?
3. Are the parameters of the pre-trained encoders (e.g., CLIP, BioBERT) included in the reported model size of 4.52M? If not, could the authors provide the total parameter count including these encoders to better assess the model's practical lightweightness?
4. In the ablation study, when CPRFL or LTFR is removed, how is the feature fusion handled in their absence? Were these modules simply replaced with naive concatenation or baseline fusion strategies?
5. In Fig. 1, what is the meaning of the pipeline “Image->Noise Image->Noise Prediction”?
6. Can you provide the pseudo code of training?

---

### Official Review · Reviewer_zBT5 · 2025-11-02

**Soundness:** 2
**Presentation:** 2
**Contribution:** 2
**Rating:** 2
**Confidence:** 5

**Summary:**

The paper proposed a prompting network for medical image segmentation. It injects text–image prompts into a diffusion-U-Net and fuses multi-scale text/vision features via an MLP-based module. Experiments are done across diverse medical imaging modalities, including CT/histopotology/X-ray.

**Strengths:**

- The paper is well-written and easy to follow.

- The experiments are done across multiple medical imaging modalities to prove the generalizability of the work.

- Source code of the paper is available in an anonymised repository.

**Weaknesses:**

- It's not clear where the text pairings for the datasets are coming from, specifically for the MoNuSeg and QaTa-COVID19 datasets. Moreover, are they fixed texts for all the images in the dataset? Or are they image-specific? If they are fixed, how do you ensure they align with the image?

- The datasets used are very small-scale. MoNuSeg uses only 5 training images (14 test), QaTa-COVID19: 150 train / 50 test from a dataset with 121k+ images and MosMedData+: 150 train / 30 test. This is a drastic sub-sampling and may dramatically affect the perforamnce of the method.

- I see the paper is not a novel contribution. It positions CPRFL/LTFR as novel, but several recent works condition diffusion/UNet with language or cross-modal adapters.

**Questions:**

- Could you elaborate on how you derived the text inputs for each modality?

- Why are the sampled datasets very few for the test set? How do you ensure the generalizability and performance of your method based on such small-scale data?

- When investigating your code in the link, the files can not be opened! Can you ensure the presence of the final, and if the access is open?

- Can you clarify on architectural contribution of the method?

**Details Of Ethics Concerns:**

N/A.

---

### Note · Authors · 2025-11-12

I have read and agree with the venue's withdrawal policy on behalf of myself and my co-authors.